# Patients' perceptions of home-based imaging diagnostics: A qualitative study

Lorena Jorge Lorenzi[1]*, Giovana Fondato Costa[1], Helianthe Kort[2], Paula Costa Castro[1,3]

1 Programa de Pós-graduação Interunidades em Bioengenharia, Universidade de São Paulo, São Carlos, São Paulo, Brazil, 2 Building Healthy Environments for Future Users, Eindhoven University of Technology, Eindhoven, Netherlands, 3 Department of Gerontology, Federal University of São Carlos, São Carlos, São Paulo, Brazil

* lololorenzi12@gmail.com

## Abstract

Expanding access to diagnostic services through home-based imaging represents a promising strategy to reduce barriers to healthcare, particularly by eliminating the need for patient travel. For this model to be viable, diagnostic equipment must be portable, and its success depends in part on patient acceptance. Despite the growing implementation of home imaging services, there is a lack of evidence in the literature regarding patient perceptions of this modality across different exam types. Therefore, the present study aimed to assess the perceptions of patients who underwent home-based imaging diagnostics. This qualitative interview study was conducted with patients who underwent home-based imaging diagnosis. The interviews were conducted by phone between May and July 2024, with up to three contact attempts. When necessary, a proxy answered on behalf of the patient. Aspects such as positive and negative points, reliability, preference, preparation, and quality of care were analyzed. The interviews were recorded, transcribed, and analyzed using inductive analysis. Thirty-four patients and their proxies participated in the study, with the majority of patients being older adults. Four types of imaging exams were performed in the home setting. Participants' perceptions were predominantly positive, emphasizing the convenience of avoiding travel, the high quality of care provided, confidence in the accuracy of the exams, and the quality of the equipment used. However, some concerns emerged, such as fear in receiving professionals at home, the need for faster result access, and better home infrastructure for exams. The insights gained from this study provide valuable guidance for the development of regulatory frameworks and operational strategies for home-based imaging services. By addressing the identified needs and concerns of end users, these findings can support the improvement of public policies, while also informing on how to effectively plan and manage such services to enhance satisfaction, adherence, and broader implementation.

**Data availability statement:** All relevant data are within the manuscript and its Supporting Information files.

**Funding:** This study was financed, in part, by the São Paulo Research Foundation (FAPESP), Brasil. Process Number 2023/05218-6. The study was also financed by the Coordination for the Improvement of Higher Education Personnel - Code 001.

**Competing interests:** NO authors have competing interests.

## Introduction

Diagnosing a health condition is essential to guide appropriate treatment. According to the Centers for Disease Control and Prevention [1], diagnosis can be defined as "the act or process of identifying or determining the nature and cause of a disease or injury through evaluation of patient history, examination of a patient, and review of laboratory data." Health diagnoses can be obtained through various methods and different types of equipment, and among the most commonly used are imaging exams such as X-ray, ultrasound, computed tomography, and magnetic resonance imaging. Imaging diagnoses can identify a wide variety of health conditions, including stroke, thyroid disorders, appendicitis, coronary artery disease, among others [2].

Detailed information about the disease and its stage allows for the selection of the most effective therapeutic approach and helps patients understand their prognosis. This enables more effective care planning, in collaboration with family members and healthcare professionals. Notably, the earlier and faster the diagnosis, the lower the risk of disease progression and complications [3,4]. Knowing one's diagnosis can support lifestyle changes, such as diet modifications, help reduce hospital stays, and prevent inappropriate medication use, ultimately lowering healthcare costs and improving outcomes [4].

Access to health care is related to the opportunity to seek, obtain, and utilize health services, as well as to have the patient's needs adequately met. Various factors may influence this access, including patient-specific characteristics, such as region of residence, socioeconomic status, and acceptability, as well as characteristics of the services, such as cost, location, adequacy, among others [5]. One strategy to improve access to diagnostics is to offer them at the patient's home, eliminating the need for travel. Home-based care particularly benefits older adults, individuals with mobility limitations, people living in remote areas, and those deprived of liberty, thereby promoting equitable access to healthcare services [6–8].

In order to perform home health diagnostics, such as imaging diagnostics, the equipment must be portable so that it can be transported to the home. An example of this type of portable diagnostic equipment was mentioned in the study by Kjelle, Lysdahl and Olerud [9], which evaluated the impact of mobile radiography services in older people's homes and identified that older people in these homes had fewer radiographs compared to the general population. However, despite this fact, in places where mobile radiography services exist, there is an increase in the number of diagnoses performed by these older people, reaching a rate closer to the rate for the total population. Thus, the importance of having home imaging diagnostic services is evident in order to increase access for people in vulnerable situations or with reduced mobility.

In addition to the studies mentioned above, a scoping review conducted by Toppenberg et al. [10], found that mobile X-rays could be implemented in homes, long-term care facilities, and shelters. They reported that both patients and healthcare professionals were satisfied with the service. The authors also noted that the image quality was adequate and that this approach may be cost-effective. Supporting this,

Nitschke et al. [11] demonstrated that portable dental X-ray equipment produced images comparable in quality to those obtained with wall-mounted devices in clinical settings.

The relevance of mobile diagnostic imaging gained further prominence during the COVID-19 pandemic. Providing radiography services at home helped reduce the risk of infection and reinforced the role of community-based medicine [12]. Although these services expanded significantly during the pandemic, they continue to be used and are growing in scope due to their demonstrated value in routine care. To ensure equitable access, it is important to optimize these services. Understanding patients' perceptions is a key step toward identifying their needs and values, enhancing service quality, and promoting patient-centered healthcare [13,14].

Patients' perceptions can be shaped by factors such as the social context, prior experiences with healthcare systems, personal expectations, and characteristics of the health system itself [13]. Therefore, these perceptions must be carefully assessed, as understanding how patients evaluate the service is fundamental for identifying the aspects they consider most valuable. Such insights not only guide institutional improvements and allow verification of ongoing enhancements but also contribute to the provision of patient-centered care [13].

To understand patients' perceptions, it is important to consider Human-Centered Design, in which the focus of inquiry guiding the development of a service is the human being and their perspectives, with the aim of producing solutions that are truly usable. This approach requires identifying users' needs and examining how they interact with the service. In the healthcare field, it involves the active participation of patients and is essential for the development of health services that are human-centered and aligned with the real-life contexts of end users [15].

Methods for capturing patients' opinions include questionnaires, interviews, focus groups, and experience reports [14]. According to a review carried out by Ofili [16], qualitative approaches are particularly effective because they allow for in-depth exploration of patients' views, including critical reflections on service quality.

Through literature searches, it was identified that there are five studies on the perception of patients, caregivers or physicians regarding the performance of imaging diagnostic at home [17–20]. However, these studies report perceptions of only one type of diagnostic equipment: the X-ray. These studies were conducted with residents of nursing homes (long-term care facilities for older adults). Geographically, two studies were carried out in Australia, one in Denmark, and one in Sweden.

Given this limited scope, there is a pressing need to investigate perceptions of diverse imaging modalities conducted at home, especially in low- and middle-income countries. This would provide a broader understanding of their acceptance, feasibility, and potential for improvement. Accordingly, the present study aimed to examine the perceptions of patients who underwent home-based imaging diagnostics in Brazil.

## Materials and methods

### Study design

This study adopted a qualitative approach and was conducted through semi-structured interviews with patients who underwent imaging diagnostics at home. The study followed the "Consolidated criteria for reporting qualitative research (COREQ)" checklist [21].

### Recruitment of participants

To assess perceptions, telephone calls were made to patients who had undergone diagnostic imaging at home through a Brazilian company located in the state of São Paulo. This company offers home-based diagnostic imaging such as echocardiograms, X-rays, ultrasounds and doppler. The calls took place between May and July 2024.

All patients who received home imaging diagnostics from one company provider in March 2024 were contacted, with up to three call attempts per patient. During the call, participants were informed about the study and their voluntary

participation. Key points from the Free and Informed Consent Form (FICF), such as data confidentiality and the right to withdraw at any time, were also explained. With authorization, the call was recorded for later transcription and analysis.

All interviews were conducted by author LJL, who at the time was a doctoral student with a bachelor's and a master's degree in gerontology, and prior experience with interviews and qualitative research. The interviewer had no previous contact with the participants. At the start of each interview, she introduced herself by name, current academic status (doctoral student), and the university conducting the research.

### Selection of participants

The study included patients who had undergone any imaging diagnostics at home, agreed to participate in the interview, and were private clients, regardless of their insurance status. This criterion ensured the assessment focused solely on the diagnostic service. Participants had to be 18 years or older. Individuals who declined to receive the informed consent form were excluded. In cases where patients could not respond, a proxy respondent—such as a formal or informal caregiver who was present during the home examination—was allowed to answer on their behalf

### Study outcomes

The outcomes were assessed through semi-structured interviews conducted via telephone calls, with open-ended questions, which were tested by the authors themselves in meetings, according to a script. Sociodemographic information on age and gender identity was collected, in addition to the type of examination performed. Participants' preferences for having the examination at home versus in a clinic or hospital were explored, along with the reasons behind their choices. The level of ease in preparing for the exam (easy, normal, or difficult) was also addressed.

Positive and negative aspects of the home exam were identified, including concerns about the procedure. Reliability of the exam and participants' opinions regarding the results—compared to those from clinics or hospitals—were evaluated. The quality of care provided by professionals was assessed in terms of punctuality, appearance, and attention to patients' questions. Finally, the participants shared their opinions on the condition and modernity of the equipment, the condition of the healthcare team's transport vehicle and suggestions for service improvements. The interview questions were based on topics from "SERVQUAL: A multiple -item scale for measuring consumer perceptions of service quality" [22]. The full interview script is available in Supplementary Material S1 File.

The perceptions examined in this study refer to participants' experiences with receiving diagnostic imaging services in the home setting. These perceptions encompassed aspects related to service quality, as well as the views and impressions of individuals who underwent home-based imaging procedures. The study did not explore perceptions associated with any particular disease or with a single diagnostic category. Instead, it adopted a comprehensive approach, focusing on the home-based diagnostic imaging service as a whole, which includes various types of examinations performed at home, such as X-rays and ultrasounds.

### Ethical aspects

This study was approved by the Ethics Committee of the Federal University of São Carlos, registered under Number: 69013223.2.0000.5504, with approval report Number: 6,454,104. All participants received the FICF, and patient data were kept confidential. Verbal consent to participate in the study was obtained during the phone call, after the research and the FICF had been clearly explained.

### Data analysis

The data from the telephone interviews were recorded as audio files, transcribed, and double-checked by two authors using Word and Excel 2016. The analysis followed the inductive methodology [23] using NVIVO 15 software. Themes

emerged directly from the data through a coding tree developed by the authors, organizing content based on thematic similarities. Two authors conducted independent analyses and resolved any disagreements collaboratively.

## Results

### Characterization of participants

Among the 140 contacts of patients who underwent home imaging diagnostic service in March 2024, 24 were duplicates. Thus, calls were made to 116 patients. After the calls, 26 were excluded, as nine patients passed away, eight patients and proxies didn't want to participate in the survey, six proxies reported that there was no one who was present on the day of the exam and would be able to answer, one gave up responding at the beginning of the questionnaire, a telephone number was wrong or the patient changed their number and one answered the questionnaire but did not want to receive the FICF and did not authorize the recording of the call, so he was excluded. In addition, it was not possible to contact 55 patients, even after three contact attempts, as there was no response or the telephone(s) contacted were only in voice-mail. The participant recruitment process is described in the flowchart in Fig 1.

Thirty four respondents participated in the study, one of whom was a caregiver for two patients who underwent diagnostic imaging services at home. The telephone calls lasted a minimum of four minutes and 17 seconds and a maximum of 23 minutes and nine seconds, with an average duration of seven minutes and 22 seconds. During the call, the interviewees were at the patient's home or at their workplace.

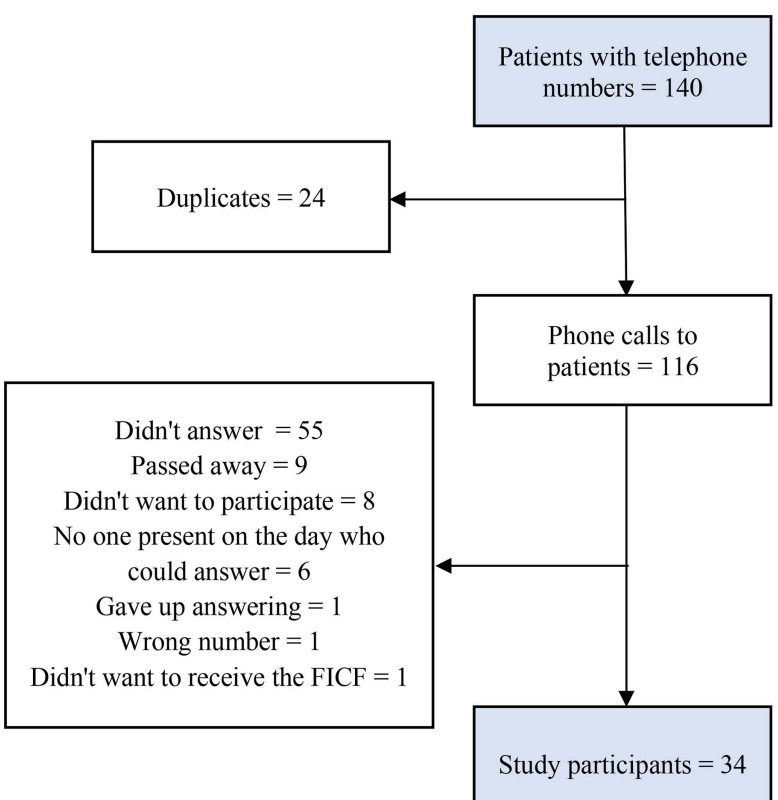

**Fig 1. Flowchart of the study participant recruitment process. Caption: FICF: Free and Informed Consent Form.** Source: Developed by the authors (2025).

During the phone calls, since many patients were unable to respond due to speech or hearing difficulties, another individual who was present on the day of the home examination(s) could respond for the patient and was called a proxy. Table 1 presents the characteristics of the study participants, including identification, age, sex, whether it was the patient or a proxy who responded, and if it was a proxy, what type of proxy.

The age of the participants ranged from 24 to 93 years, with a mean of 53.9 (±14.5) years. The majority (82.3%) were female and were a proxy for the patient (91.2%). The proxies were formal caregivers, spouses, children, parents, nieces, and one did not report. The proxies responded about the home diagnosis performed on patients who were three

**Table 1. Characterization of research participants.**

| ID | Age | Sex | Respondent | Proxy Type | Patient' Age |
|---|---|---|---|---|---|
| P_01 | 70 | Masculine | Proxy | Son | 90 |
| P_02 | 62 | Feminine | Proxy | Daughter | 91 |
| P_03 | 61 | Feminine | Proxy | Did not report | 90 |
| P_04 | 50 | Feminine | Proxy | Daughter | 93 |
| P_05 | 46 | Feminine | Proxy | Daughter | 77 |
| P_06 | 66 | Feminine | Proxy | Daughter | 92 |
| P_07 | 53 | Feminine | Proxy for 2 | Caregiver | 94 and 96 |
| P_08 | 64 | Feminine | Proxy | Wife | 65 |
| P_09 | 52 | Masculine | Proxy | Son | 82 |
| P_10 | 61 | Feminine | Proxy | Daughter | 85 |
| P_11 | 41 | Feminine | Proxy | Niece | 86 |
| P_12 | 42 | Masculine | Proxy | Caregiver | 75 |
| P_13 | 24 | Feminine | Proxy | Mother | 2 |
| P_14 | 60 | Feminine | Proxy | Daughter | 95 |
| P_15 | 60 | Feminine | Proxy | Wife | 94 |
| P_16 | 31 | Feminine | Proxy | Mother | 11 |
| P_17 | 34 | Feminine | Proxy | Caregiver | 79 |
| P_18 | 48 | Feminine | Proxy | Caregiver | Did not report |
| P_19 | 49 | Masculine | Proxy | Caregiver | 94 |
| P_20 | 93 | Feminine | Patient | No proxy (patient) | 93 |
| P_21 | 55 | Masculine | Proxy | Caregiver | 91 |
| P_22 | 71 | Feminine | Proxy | Caregiver | 78 |
| P_23 | 61 | Feminine | Proxy | Wife | 80 |
| P_24 | 53 | Feminine | Proxy | Daughter | 85 |
| P_25 | 53 | Feminine | Patient | No proxy (patient) | 53 |
| P_26 | 40 | Feminine | Proxy | Daughter | 70 |
| P_27 | 57 | Feminine | Proxy | Daughter | 87 |
| P_28 | 35 | Feminine | Patient | No proxy (patient) | 35 |
| P_29 | 85 | Masculine | Proxy | Husband | 82 |
| P_30 | 45 | Feminine | Proxy | Caregiver | 88 |
| P_31 | 41 | Feminine | Proxy | Mother | 3 |
| P_32 | 63 | Feminine | Proxy | Daughter | 87 |
| P_33 | 43 | Feminine | Proxy | Mother | 2 |
| P_34 | 64 | Feminine | Proxy | Daughter | 96 |

Legend: ID: Identification.

Source: Developed by the authors (2025).

newborns, one child, two adults and the others were all over 65 years old. Only three patients participated in the study, all of whom were female and aged 35, 53 and 93 years.

### Answers about performing diagnostic imaging at home

During the interview, participants answered 12 questions about performing the diagnostic imaging service at home. The themes of the questions asked were allocated as the category names in the analysis, being: Type(s) of exam(s), Preference for examination location, Preparation for the exam, Positive aspects of performing the exam at home, Negative aspects of performing the exam at home, Fear of performing the exam at home, Confidence in the exam performed, Difference between the exam and the clinic/hospital exam, Professional service, Opinion on equipment used, Opinion on transportation vehicle, Suggestion for changes in the service. Supplementary Material 2 in S1 File presents the codebook with the categories, subcategories, their descriptions and the number of files and references found in each one.

### Type(s) of exam(s)

Regarding the type of diagnostic imaging exam that the patient has undergone at home, the participants answered four different exams: x-ray (mentioned by 79.4%), ultrasound, echocardiogram and doppler echocardiogram. In addition, participants reported other types of tests they performed at home, such as electrocardiogram, electroencephalogram, Holter, ABPM. Some patients underwent only one type of exam, but others have undergone several types.

*"Oh, she has already had an ultrasound, an x-ray. These imaging tests... almost all of them, like that. She has ultrasounds and x-rays frequently, carotid ultrasounds, you know, lung x-rays." (P_ 11)*

*"Yeah, well, he already had an X-ray, he had a doppler echo and he had an echocardiogram, those were the three he did." (P_15)*

### Preference for examination location

Regarding the preference for the location of the exam, most participants (91.2%) reported that they prefer the patients to take the exam at home, and the reasons for preferring home were allocated into the following subcategories: transportation cost, due to the cost of having to move the patient from home to a clinic; difficulty in mobility, in cases where the patient has difficulty moving; avoiding disruptions to the routine and schedule, since doing the exam at home does not require changing the routine of the patient and those responsible for moving the patient; practicality, due to the convenience of doing it at home; and avoiding the risk of infection, as it can help reduce the risk of infection if the exam had to be performed at a clinic or hospital.

*"So, it is essential, right, for someone who is bedridden. She is 91 years old and has been bedridden for over 10 years, it is... transportation to a clinic, to a hospital is not feasible because of the ambulance. These are patients who feel a lot of pain, which does not help at all in moving their bodies, in addition to all the... the stress, right [...]" (P_02)*

*"[...]We don't have the availability of an ambulance through the insurance plan we have. So, we have to pay for it and it's not cheap [...]" (P_14)*

*"My daughter is... she is special, and I prefer to be home-based because that way I don't have to go out with her or put her in... in an environment where she could be at risk of contamination... and I also don't have to move her around a lot." (P_33)*

*"[...] I didn't have the availability to go to a... um... how do you say it? A place, I was very busy with work, and that made things much easier for me, because I didn't need to, um... move, neither she nor I, so it was wonderful." (P_24)*

However, three participants reported that they prefer the exam to be performed in a clinic, due to the reduced space at home for using portable diagnostic equipment and because they prefer to go in person to a clinic to perform the exam.

*"Well, I usually go to the clinic, right? (Do you prefer a clinic?) That's it. (Why do you prefer a clinic?) Oh, I don't know, I think it's just that here at home, sometimes there's... there's not much... how do you say it? Space, right? To go through with the machine. All that stuff. When you can, right." (P_25)*

*"Girl, I... I did it here, because I was, I needed it urgently. And... at the time I couldn't, I hadn't managed to get any space at... in a physical hospital, so, that's why I scheduled it here, but... for me I prefer to go in person." (P_16)*

### Preparation for the exam

Regarding patient preparation for the examination at home, the responses were grouped into three subcategories: easy preparation, reported by approximately half of the participants that preparation for the examination was done easily and calmly; no preparation, in cases where there was no need to prepare for the examination, such as for X-rays, for example; and similar to the hospital-clinic, reporting that preparation is the same as if the examination had been done in a clinic or hospital.

*"Calm, they go through it properly, no worries." (P_11)*

*"No, it didn't require any preparation." (P_31)*

*"[...] she has bowel incontinence, depending on the day and time, sometimes more, sometimes less, sometimes more when she is not taking antibiotics, sometimes much less incontinence when she is taking antibiotics. So all of this interferes. Yes... but we try to give her the bottle of water, very, very close to the time the doctor will arrive to do the exam and with that we have been able to do it. But this would happen in a clinic or even in a hospital. And it is not difficult because it is a matter for her, as a patient." (P_14)*

### Positive aspects of performing the exam at home

Regarding the positive aspects of performing the diagnostic imaging exam at home, some subcategories were created, such as access to the result (images), about having access to the exam results after they are performed; flexible schedule, due to the exam being scheduled urgently and having flexibility of schedule according to the patient's needs; exam quality, reporting that the quality of the exam performed is good; speed of service execution, about the speed and agility in performing the service at home; and speed of the result, about the speed in obtaining the exam result after it is performed.

*"The practicality, right? You receive the image, you forward it to the geriatrician and he/she already gives you the procedure." (P_21)*

*"Accessibility, fast, it's fast, it arrives fast. It's... there are several, several time options, it's not like a clinic where... it's not available at all times, it's never at the same time that we want." (P_16)*

*"[...] Quality too, I think it's good quality." (P_17)*

*"They were... Agile, they were fast, you know? They did it really quickly[...]" (P_10)*

*"The results come out practically immediately, right? For example, the X-ray, which is what is done most frequently, the result comes out of the technician's computer, the technician sends it to the home care, the doctor approves it and says yes, says no, and sees what's wrong." (P_13)*

Other subcategories identified as positive points of performing the diagnostic imaging exam at home were the absence of travel, as there is no need to travel to perform the exam; comfort of doing it at home, the comfort and convenience of being able to perform the exam at home; and avoid the risk of infection, since performing the exam at home avoids the risk of the patient contracting an infection in a clinic or hospital.

*"The main thing is not having to take her to a hospital, unnecessarily." (P_02)*

*"Wow, everything, practicality and comfort.... yeah... yeah... for an older person it's really good to have this comfort of doing things at home, taking exams at home." (P_04)*

*"[…] That's not the case at the time, but even more so in this cold season, I think... to avoid putting them in... the lab, in clinics with a lot of people, especially with these... with all these flus, with all these things [...]" (P_24)*

In addition, four other subcategories were identified regarding the positive aspects of performing the exam at home, namely: notice regarding the performance of the service, reporting that company called before going to perform the service, confirming whether it was possible to go; patient care, regarding the care that the professionals have with the patient during the exam; healthcare team punctuality, due to the fact that the professionals are punctual and arrive at the home on time; and the quality of service, reporting points that they liked about the service provided by the professionals who went to perform the exam, such as being attentive, professional, kind, polite and helpful.

*"They even had a positive point, which is, they always called before coming to the residence, so they called saying, look, can I go and so on?" (P_02)*

*"[...]The second point is that there was a lot of punctuality when it came to do the exam and the people who came to do the exam were very careful because of her frailty." (P_01)*

*"The service was very positive, it's... the people in this specific laboratory, when they were here, were very helpful, the service is very humanized." (P_14)*

**Negative aspects of performing the exam at home**

The negative aspects of performing diagnostic imaging at home were grouped into a few categories, namely: delay due to traffic, one person reported a delay in professionals arriving at the home due to city traffic; delay in scheduling the service, regarding the difficulty and delay in scheduling the service for a close date; and home infrastructure for carrying out the exam, reporting that the home infrastructure is a problem for performing the exam, due to the space in the rooms for the size of the diagnostic equipment and the electrical grid to support their power.

*"(Was there any negative point?) No, just... just, the time was a little late due to traffic, that's all." (P_15)*

*"It was good, it was great, but I thought... it was just that I thought it took a long time to get an appointment [...] I think it took almost a month, 20 days, to get an appointment and he needed it urgently." (P_23)*

*"[...]It was some kind of equipment, I think it's very high-tech, I'm not sure, and it needed a very powerful electrical system. My mom lives at the back of my house, with a room and kitchen I built for her, and the electrical installation there wasn't... it's not really up to the company's standards, you know? And with such powerful equipment, it would end up cutting the power, but they were able to do it[...]" (P_02)*

Other negative points mentioned were: delay in accessing the results, with participants reporting that they would like to have access to the test result at the time the test is being performed; exam report with summarized information, regarding the exam report containing little information; exam report not sent, due to the fact that the exam report was not sent and the proxy had to request it from the responsible company; and the cost of home exam, due to the amount charged for the service to be performed at home. The Brazilian public Unified Health System (SUS) to date does not provide coverage for this examination. Generally, the procedure is commissioned by private health insurance plans, and when obtained on a private basis, its cost is not reimbursed. In the context of this study, all participants procured the service privately and, consequently, assumed full financial responsibility for the examination. Most participants reported no negative points, as they did not identify any negative aspects.

> "[...] I missed that at the time, not afterwards, at the time of the exam I could have access, but afterwards, they put the image on the platform and put the... the exam report as well. (Would you like to have access at the time, you say?) It's... sometimes at the time, perhaps due to an emergency situation, we can get and pass this image, this report, let's suppose in a faster way to the doctor so that he/she can be in a position to make a decision." (P_14)

> "[...] I think the exam report could be more complete. The only thing I think is, you know, nothing more, it's just my opinion, then I had to take the exam to the neurologist for evaluation, but the neurologist himself said: well, I'm going to look at the map itself, not the report, because the report is very summarized, it doesn't have the observations. [...] In the first case the report wasn't sent, I had to ask for it [...]" (P_26)

> "No, it would be more the value (cost), right? But like... I understand because of... How it's done, right? There's no other way, then." (P_05)

"No, nothing negative." (P_04)

**Fear of performing the exam at home**

Regarding fear of having the exam performed at home and having the healthcare team go there, the following subcategories were verified: radiation concerns due to the concern of patients and proxies about radiation protection for performing the x-ray; fear of deception in identification, as some patients and proxies report a concern of being deceived by other people pretending to be health professionals; fear during the COVID-19 pandemic, reporting fear of professionals coming at the end of the day or at night during the COVID-19 period, due to the fear of them being infected; and no fear, with the majority of participants reporting that they were not afraid of the exam being performed at home and the healthcare team going there.

> "Yes, obviously, X-rays are always very dangerous, right? They provide some protection, but it's never the same as doing it in an appropriate place, with lead protection, etc., which are necessary, but it's... in emergencies, sporadically, I see it as, as good." (P_29)

> "We always ask for identification first, right? When the person arrives, they introduce themselves. What can you do? In any case, we take risks. There are good professionals, but there are also a lot of people who deceive us, right? But every time I needed someone, they asked for it, they sent the name of the person who was coming, they identified themselves, no problem." (P_06)

> "Yes, actually, during the COVID period, I had, because of COVID, not because of the professional, I already knew all the professionals, they were all like that... ethical, polite, no, no, I wasn't afraid, besides the fact that they come with the car, with the identification that belongs to the company, they present themselves with a company badge, so I wasn't

*afraid, what was scary was that in the afternoon they came late at night, during the COVID period, which was when they most... I don't know, I think one of the last times they came, I was afraid of the sanitization of the equipment if they came late at night, but that's it, not the rest." (P_02)*

*"None, no, none at all." (P_26)*

### Confidence in the exam performed

Regarding the confidence that patients and proxies have in the examination that was performed at home, the subcategories found in the responses were: trust the exam, reported by more than half of the participants who mentioned that they trust the diagnostic imaging examination that was performed at home; depending on the complexity of the diagnosis, reporting that they trust the diagnostic imaging examination performed at home depending on the complexity of the diagnosis and that these portable equipment are better for simpler and more urgent diagnoses; depending on the patient's situation, reporting that performing the diagnostic imaging examination on a bedridden patient may interfere with the image, but the participants reported that there was no problem with the image; uncertainty depending on the type of exam, with the participants reporting that they have confidence in the diagnostic imaging examination they performed, for example an x-ray, but that they are not sure about other more complex examinations, such as an ultrasound or magnetic resonance imaging;

*"For some types of... use, wow, it's great, you don't need more than that... So it depends, it depends, right, if it's a diagnosis, something more complicated, then you need a... more powerful equipment to see the images clearer, right, but for some simpler cases, wow, it perfectly solves the need." (P_04)*

*"No... we didn't notice anything bad about the image, we know that to take an image of a bedridden patient it is... anyway, in home care, you need to have the positioning… you need to have the positioning of the patient, we are not in the field, but we are learning a lot from this situation at home, so we know that there are things that can interfere, but the times we had the images enough for a check." (P_14)*

*"[...] I do trust, because mainly it was more like... x-rays, it was more like x-rays, right? I know it seems like they do ultrasounds too. So, we never used ultrasounds. I would perhaps be a little afraid of ultrasounds, because it would need to have a doctor come in to do the ultrasound, which is different from the x-ray image, so maybe an ultrasound, as I never did with them...I never needed it." (P_02)*

In addition, other categories mentioned regarding trust in relation to the diagnostic imaging exam performed and its result were: professional qualification, regarding the positive influence that the qualification and service of the professionals have on trust regarding the exam; exam well done, regarding trust in the result because the exam was well done and not performed quickly; and trust due to medical opinion, several participants (approximately 41.2%) reported that they trust the exam result because the patient's doctor checked the exam and there was no problem.

*"I think that... given the professional capacity that I saw here, the professionalism, I think that you can trust him, very serious, focused on what he/she was doing, so... you can trust." (P_22)*

*"And the guy did a great job, it took a long time. In the health plan, when we get there, it's fast... five... it doesn't even last three minutes, already to... put the device there, I don't know what the hell that is. And at home it was done very well, because they did it and redid it." (P_15)*

*"Look, in her case, an x-ray was taken, right? And it was sent to the doctor for analysis. So we got a very, very quick response. Including the issue of the x-ray, I never had a problem with...of giving something, a result... that's divergent, something like that." (P_33)*

### Difference between the exam and the clinic/hospital exam

Regarding the difference between a diagnostic imaging exam performed at home and one performed at a clinic or hospital, this topic was addressed in the question about confidence in the exam that was performed and it was the only question that did not receive answers from all participants, being answered by 82.4%. The subcategories found were: images with lower quality than those in the clinic, reported by three people who believe that the equipment used at home does not have as much sharpness and quality as those at the clinic; depends on the quality of the clinic, mentioning that the difference depends on the quality of the clinic being compared; doesn't know how to evaluate; two participants did not know how to evaluate whether the exam performed at home is the same as one performed at a clinic; and same – no difference, with the majority of respondents (75%) believing that there is no difference between a diagnostic imaging exam performed at home and one performed at a clinic or hospital.

*"Yeah... I, I think he's not as qualified, let's say, as you do in a clinic that has much more precision, but as an indication, yeah, very close to the desired effect this... this happens. I think it can, to, to help a lot in these special moments." (P_29)*

*"Oh, I can't tell you that, it's not that simple. It depends on the clinic, it depends on the place, right? It depends... I was satisfied with the service, I can tell you that, right? I can't compare because it depends on the clinic." (P_01)*

*"I don't know how to evaluate. So [...] in reality, I like tomography so not always the X-ray, it points out... I don't know, I could be wrong... so, if there is any discomfort, something, we go to the X-ray first, I don't know if there is a difference in a clinic, maybe the resolution... it's... I don't know, I don't know how to evaluate that, but their resolution is very good, okay? [...]" (P_27)*

*"No, there is no difference, it is the same. The best thing is the comfort for the patient, in her case, in her condition, right? It is much more comfortable for her." (P_18)*

### Professional service

Regarding the care provided by professionals during the diagnostic imaging exam at home, the vast majority of participants reported good service from professionals and that they were attentive, punctual, and equipped. In addition, the following other subcategories were found in the responses: image check, due to the health professionals checking with a doctor to see if the image is good before leaving the home; behavioral skills of professionals, regarding the patience and manner the professionals have with the patient, who in this case was a child; hygiene, regarding the hygiene and use of masks by professionals, such as during the COVID-19 pandemic; qualification of professionals, regarding the professionals being trained to perform the service; and a negative subcategory mentioned was punctuality problem, reported by a participant who mentioned that in one of the exams performed there was a delay in performing the service due to a problem with the company vehicle, but she reported that she was notified and that there was no problem.

*"[...]They wait too, right? Sometimes the doctor, for example, when there is a change of nasoenteral tube, then they need the doctor's approval before they can leave. So, they wait, very calm, right? There is no... problem, you know?" (P_13)*

*"They arrived at the correct time, they have. They managed to be good... how can I tell you? They have a lot of patience with children too, he has a lot of ways with children, both times they went I found that."* (P_31)

*"I remember that at the time of the pandemic, they had a lot of safety criteria, like they came with a mask, they... they sanitized all the equipment with alcohol, they... they had some really cool protocols even because of the COVID pandemic."* (P_02)

*"I found them to be highly capable."* (P_15)

*"[...]I understand there was a problem with the vehicle, the vehicle... Anyway. She said, no problem, that's what I 'm saying, the person is at home, you know, is she waiting? No problem, it's not like in a clinic where you have to wait there, right? But at home... it's nothing to discredit the issue of the delay [...]"* (P_26)

### Opinion on equipment used

Regarding the participants' opinion on the equipment used to perform the diagnostic imaging exam at home, all reported that it was in good condition and the majority reported that it appeared to be modern, although some (17.7%) did not know how to report on its modernity.

*"Yes, they are modern, it is... That's the question I always ask around here, the equipment is modern and in perfect condition."* (P_12)

*"Look, I have no idea about the modernity, but it was in good condition."* (P_09)

In addition, four other subcategories were identified regarding the diagnostic imaging equipment, namely: equipment portability, mentioning that the equipment currently used is smaller and portable; equipment quality, reporting that the image quality is good and that the equipment is up to date; and high electrical load of the equipment, which appeared negatively, as one participant reported that due to the high electrical load of this equipment, the home's electrical network could not handle it.

*"It is in good condition, portable and easy to handle."* (P_21)

*"When she started using x-rays at home, it was a little different, it seemed like that because she was in bed, atrophied, it was... it looked like it had a foot, it was a bit complicated, now this one that looks like a camera, right, that he's holding right there, it's easier. I prefer it."* (P_32)

*"[...] the equipment was always good, I think too good, they were big, I think maybe with more advanced technology and even gave better image performance."* (P_02)

*"[...] in the case of my house here, which is a simple house, it's... the electrical installation couldn't always handle, I don't know... the load, you know?"* (P_02)

### Opinion on transport vehicle

Regarding the participants' opinion about the transport vehicle that was used by the professionals to go to the home and take the diagnostic imaging examination equipment and the professionals, half reported not having seen the vehicle, for example due to the fact that they live in an apartment, and the other participants saw the vehicle and said that it was in good condition, but one said he did not remember what the vehicle was like.

*"No, I didn't, because we're in an apartment, I didn't... get to see it."* (P_27)

*"Good vehicle, yes, yes, stickered, new vehicle. I didn't see any... I personally didn't see any problems."* (P_14)

*"I don't remember."* (P_23)

**Suggestion for change in the service**

Participants were asked if they had any suggestions for changes in the diagnostic imaging service performed in their homes. The vast majority (76.5%) reported that they had no suggestions for changes. In addition, other subcategories found in the responses were: access to exam results (images), due to delays in access or lack of access to exam results; review of the exam report with the perspective of different professionals, suggesting that the exam report be reviewed by professionals from different specialties to make a more assertive diagnosis; and delay in obtaining the service, mentioning that in some cases, such as emergencies, it takes a while to schedule the service.

*"No, nothing, everything is fine. So far I've never had any problems and I don't think I will."* (P_32)

*"If they could make the image available to family members, right? because then it falls into home care... goodbye, right? if you have another doctor, you don't have the images [...]"* (P_02)

*"No, just a report, because from what I understand, it seems that the exams are done and the images are sent to a doctor for a report, from what I understand, that's it, isn't that it? That's why sometimes it takes a while, I don't know what, anyway, my only point is just that."* (P_26)

*"[...] So, maybe you know, if I could suggest, maybe a radiologist's report and in some cases, a confirmation from a physiotherapist, for example, if it's a pulmonary condition, for example, I don't know. And a more specialized doctor, a review, a joint look, I don't know... because in one of those cases, if I didn't have this, this... let's say, this assistance opinion from the physiotherapist and if we, as a family, had only seen the... the pleural effusion and had no way of passing the image to the doctor, this is very important, the platform, it is... at least it allows this, right? maybe we wouldn't have seen the pneumonia [...]"* (P_14)

*"[...] I think there was a little bit... But that's understandable, right? It's... Not always, like there was... It was always in an emergency situation, sometimes you couldn't do it right away, let's say... But the time was still a reasonable amount of time, right?... Nothing, like... also so... disparate, you know?[...]"* (P_01)

The following categories were also mentioned as suggestions for changes in diagnostic imaging services: reduce the cost of the service, as it was reported that the service is expensive and one individual suggested reducing the monetary value; healthcare team composed of men and women; suggesting that the teams of health professionals who go to the home to perform the exam be composed of men and women, in pairs, for example, so that a man does not go alone to a house of a woman; choosing the location for the exam, about letting the patient or proxy choose the place in the home where they prefer to perform the exam; and scheduled time in advance, about having a time scheduled in advance for the exam.

*"Ah, if there's a way to make it cheaper, I'd appreciate it... Because the service is a little expensive, but I understand... The system and everything that happens, right?... In relation to that. But... The only... the only point would be the prices, but, like... You have to do it, you go after it, right?"* (P_10)

*"Maybe, maybe in my case, right? It was a man who came to see me, I don't know if it's a company procedure that it's just one person, but I believe that if it was a couple, I think I would feel a little safer, right? A couple of a man and a woman, anyway. I took the test, I called my husband to stay close, so we have this fear."* (P_28)

*"Yes, the suggestion: she did it on the sofa, I found the sofa a bit... quite uncomfortable, inconvenient, right? For her. But... I think that's all. (A: So doing it somewhere else?) Yeah... maybe she could have done it on the bed, I don't think we thought about it at the time, right? I don't know, but that's all, it wasn't like that... but it was done and everything was fine."* (P_22)

*"So I think, like, um, if... if it could be done kind of by appointment, right?"* (P_02)

### Frequency of subcategories mentioned by participants and type of exams performed

The most mentioned subcategories by participants regarding home imaging diagnostic exams were related to their preference for undergoing the exam at home due to mobility difficulties and the ease of preparation or lack of required preparation. Additionally, the absence of travel, the quality of care, and the comfort of being able to undergo the exam at home were highlighted as positive aspects, and the most performed exams were x-rays and ultrasounds.

In addition, participants mentioned trust in the exam and in the medical opinion, the exam being equivalent to those performed in clinics or hospitals, the quality of service provided by professionals, the use of modern and well-maintained equipment, and the condition of transport vehicles, whether in good condition or not seen. Most participants reported no negative points, fear of undergoing the exam at home, or suggestions for improvement. Table 2 presents the categories, subcategories, the number of participants who mentioned each, and the types of home imaging diagnostic exams performed. This table was generated using the results of the qualitative data analysis conducted with NVivo software.

It can be observed in the table above that the high electrical load of the equipment and the negative aspect of the home infrastructure for performing the exam were mentioned by participants who underwent X-rays at home. Additionally, individuals who had X-rays also mentioned the fact that the equipment is portable, suggested being able to choose the location in the house where the exam would be performed, and one individual expressed concern about radiation from the equipment during the exam.

It was also noted that uncertainty about trusting the exam performed at home, whether due to trust depending on the type of exam or concerns that the images generated by the equipment might be of lower quality compared to those used in clinics or hospitals, was mentioned by individuals who had only undergone X-rays and ultrasounds. Additionally, the cost of having the exam at home was mentioned as a negative point and as a suggestion for reduction by individuals who underwent X-rays, echocardiograms, and ultrasounds.

## Discussion

The study included 34 respondents, consisting of three patients and 31 proxies. Among both the patients who participated in the interview and those whose responses were provided by a proxy, the majority (75.5%) were over 65 years old. Patients underwent four types of home-based imaging diagnostic exams, with X-rays being the most prevalent. Regarding participants' perceptions, the main highlights were the preference for home-based exams due to mobility difficulties and the convenience of not having to travel. The quality of care, trust in the exam, and its equivalence to those performed in clinics were considered positive aspects. Participants also mentioned the modernity of the equipment and the ease of exam preparation. Most did not report any negative points, concerns, or suggestions for improvement.

According to a literature review on qualitative study sample sizes by Kumar, [24], most studies found had between 20 and 30 research subjects. The present study included 34 respondents, which exceeds the average identified in the literature and is considered adequate to ensure depth and diversity of qualitative data.

**Table 2. Categories, subcategories, number of participants who reported and types of diagnostic imaging exams performed at home.**

| Categories | Subcategories | n | Type of examination performed at home |
|---|---|---|---|
| **Preference for examination location** | Home – Difficulty in mobility | 27 | X-ray, USG, ECHO, Doppler |
| | Home – Avoid the risk of infection | 3 | X-ray, USG |
| | Clinic (preferred) | 3 | X-ray, ECHO |
| | Home – Avoid disruptions to the routine and schedule | 2 | X-ray, USG |
| | Home – Practicality | 2 | X-ray |
| | Home – Transportation Cost | 1 | X-ray, USG |
| **Preparation for the exam** | Easy preparation | 18 | X-ray, USG, ECHO, Doppler |
| | No preparation | 15 | X-ray, USG |
| | Similar to the hospital-clinic | 1 | X-ray, USG |
| **Positive aspects of performing the exam at home** | Absence of travel | 19 | X-ray, USG, ECHO, Doppler |
| | Quality of service | 15 | X-ray, ECHO, USG, Doppler |
| | Comfort of doing it at home | 11 | X-ray, USG |
| | Speed of service execution | 8 | X-ray, USG |
| | Speed of the result | 5 | X-ray, USG |
| | Flexible schedule | 4 | X-ray |
| | Patient care | 3 | X-ray |
| | Avoid the risk of infection | 3 | X-ray, USG |
| | Healthcare team punctuality | 3 | X-ray, USG |
| | Access to the result (images) | 2 | X-ray |
| | Notice regarding the performance of the service | 1 | X-ray |
| | Exam quality | 1 | USG |
| **Negative aspects of performing the exam at home** | No negative points | 26 | X-ray, USG |
| | Home infrastructure for carrying out the exam | 2 | X-ray |
| | Delay due to traffic | 1 | X-ray, ECHO, Doppler |
| | Delay in accessing results | 1 | X-ray, USG |
| | Delay in scheduling the service | 1 | USG |
| | Exam report with summarized information | 1 | X-ray |
| | Exam report not sent | 1 | X-ray |
| | Cost of home exam | 1 | X-ray, ECHO |
| **Fear of performing the exam at home** | No fear | 25 | X-ray, USG, ECHO, Doppler |
| | Fear of deception in identification | 7 | X-ray, USG |
| | Radiation concerns | 1 | X-ray |
| | Fear during the Covid-19 pandemic | 1 | X-ray |
| **Confidence in the exam performed** | Trust the exam | 22 | X-ray, ECHO, USG |
| | Trust due to medical opinion | 14 | X-ray, USG, ECHO, Doppler |
| | Professional qualification | 4 | X-ray, USG, ECHO, Doppler |
| | Depending on the complexity of the diagnosis | 3 | X-ray, USG |
| | Uncertainty depending on the type of exam | 2 | X-ray, USG |
| | Depending on the patient's situation – bedridden | 1 | X-ray, USG |
| | Exam well done | 1 | X-ray, ECHO, Doppler |
| **Difference between the exam and the clinic/hospital exam** | Same – no difference | 21 | X-ray, ECHO, USG, Doppler |
| | Images with lower quality than those in the clinic | 3 | X-ray, USG |
| | Doesn't know how to evaluate | 2 | X-ray |
| | Depends on the quality of the clinic | 2 | X-ray, USG |

*(Continued)*

**Table 2.** (Continued)

| Categories | Subcategories | n | Type of examination performed at home |
|---|---|---|---|
| **Professional service** | Good service from professionals | 33 | X-ray, USG, ECHO, Doppler |
| | Qualification of professionals | 2 | X-ray, ECHO, Doppler |
| | Image check | 1 | X-ray, USG |
| | Behavioral skills of professionals | 1 | X-ray |
| | Hygiene | 1 | X-ray |
| | Punctuality problem | 1 | X-ray |
| **Opinion on equipment used** | Equipment in good condition | 34 | X-ray, USG, ECHO, Doppler |
| | Modern equipment | 28 | X-ray, USG, ECHO, Doppler |
| | Equipment portability | 4 | X-ray |
| | Equipment quality | 4 | X-ray, USG, ECHO, Doppler |
| | High electrical load of the equipment | 1 | X-ray |
| **Opinion on transport vehicle** | The vehicle was not seen | 17 | X-ray, USG, ECHO, Doppler |
| | The vehicle was in good condition | 16 | X-ray, USG, ECG |
| | Can't remember the condition of the vehicle | 1 | USG |
| **Suggestion for changes in the service** | No suggestions | 26 | X-ray, USG, ECHO, Doppler |
| | Access to exam results (images) | 3 | X-ray, USG |
| | Delay in obtaining the service | 2 | X-ray, USG |
| | Reduce the cost of the service | 1 | USG |
| | Healthcare team composed of men and women | 1 | X-ray |
| | Choosing the location for the exam | 1 | X-ray |
| | Scheduled time in advance | 1 | X-ray |
| | Review of the exam report with the perspective of different professionals | 1 | X-ray, USG |

Source: Developed by the authors (2025).

Legend: ECHO: Echocardiogram, USG: Ultrasonography.

The majority of patients in this study who received home-based imaging diagnostic services were older individuals. According to the World Health Organization [25], there is a growing trend in the proportion of older people in the population, with a consequent decrease in other age groups. Projections indicate that by 2030, the number of people over 60 years old will rise to 1.4 billion, reaching 2.1 billion by 2050. This highlights the importance of diagnostic services for the older population, which continues to grow.

In this context, aging can lead to biological changes, such as reduced functional capacity and the onset of geriatric diseases and syndromes [25,26]. Due to these aging-related impairments, many individuals face mobility difficulties. This may explain why the absence of travel was the most cited positive aspect regarding home-based exams and why participants preferred undergoing exams at home due to mobility challenges.

Another positive aspect mentioned by participants was avoiding the risk of infection. Some respondents stated that they preferred to have the exam at home to avoid potential exposure to diseases when visiting a hospital or clinic. The World Health Organization's report "Global Report on Infection Prevention and Control" discusses the risks of infection during hospital stays and emphasizes the importance of infection prevention and control, concerns that align with those of the patients in this study [27].

Despite the positive aspects and the fact that most participants did not mention negative points, two respondents highlighted home infrastructure as a drawback for conducting imaging exams. This was due to small room sizes in relation

to the diagnostic equipment, as well as the high electrical load required by these devices, which at times exceeded the home's electrical capacity. In this regard, the study by Chabouh et al. [28] identified the importance of strategic planning for home-based healthcare and how various challenges can impact its implementation. This underscores the need to investigate aspects related to home healthcare execution and to adequately plan healthcare services to prevent potential issues, such as those linked to home infrastructure.

Additionally, safety is another crucial factor to consider when providing healthcare services at a patient's home. It involves various aspects, including the professionals delivering the service, family members, the patient, environmental conditions, and the equipment used [29,30]. In this regard, the present study identified certain concerns among participants about professionals coming to their homes for imaging diagnostics. The main concern was the security risk of allowing strangers into their homes and the possibility of being deceived, followed by hygiene precautions during the COVID-19 pandemic and concerns about radiation exposure from the equipment. These findings highlight the importance of considering various safety aspects when conducting home-based healthcare services.

However, despite the concerns mentioned by some respondents, the majority stated that they trust home-based imaging diagnostic exams and consider their results equivalent to those performed in clinics or hospitals. A study by Kamal et al. [31] compared the quality of portable X-rays with traditional X-rays used in healthcare facilities and found that the images produced by portable equipment were of comparable quality. This suggests that both scientific research and participants' perceptions support the reliability of portable imaging diagnostic devices.

In addition to exam quality, 97.1% of participants mentioned the good service provided by professionals, and nearly half (44.1%) highlighted quality of care as a positive factor. In this context, studies by Silva, Gomes, and Maia [32] and Meneses-La-Riva, Suyo-Veja, and Fernández-Bedoya [33] emphasized the importance of humanized care in healthcare, including empathy, respect, and attentiveness. Thus, professional care quality is an essential factor in patient healthcare and should be provided at a high standard.

Given that this study was based on open-ended qualitative interviews, the themes were analyzed descriptively, according to whether each perception was present or absent in participants' perceptions. However, future studies that use structured questionnaires or mixed-methods approaches could incorporate Likert-type items to quantitatively assess the intensity of patients' perceptions. Such standardized measures would enable more detailed comparisons across different imaging exam types and provide a more granular understanding of patient acceptance.

One limitation of this study is the low representation of patients, as only three of the 34 participants were the patients themselves. The majority of responses were provided by proxies due to the patients' health conditions or age, such as in the case of newborns. However, all respondents were present during at least one home-based imaging exam, and the proxies were either formal or informal caregivers of the patients. Another limitation was that the interviews were conducted over the phone, preventing the observation of participants' nonverbal communication.

Despite these limitations, the study presents several important strengths. It is an original study, being the first to explore patient perceptions of different types of home-based imaging diagnostics. Additionally, the study's sample size allows for an in-depth analysis of participants' perceptions. Other strengths include the use of semi-structured interviews with only open-ended questions, which likely contributed to data richness and enabled a comprehensive exploration of patient perceptions. The study also provides valuable insights for the development and enhancement of home-based imaging diagnostic services.

## Conclusion

The patients were primarily older individuals, and a total of four types of imaging diagnostic exams were performed at home. Various aspects of participants' perceptions regarding home-based exams were identified, with the majority being favorable, such as the convenience of not having to travel, the quality of care, trust in the exam, and the quality of the equipment. However, although less frequently mentioned, some negative aspects or areas for improvement were noted,

such as concerns about welcoming the healthcare team into their home, the need for faster access to exam results, and the home infrastructure required for conducting the exams.

Thus, the various aspects identified in this study can contribute to improving public policies for the regulation of home-based diagnostic services, as well as assisting companies and professionals in planning and managing these services according to the needs of end users. Since the research was conducted via phone interviews, we suggest that future studies conduct in-person interviews with patients and also gather the opinions of individuals who have never used this type of service.

## Supporting information

**S1 File. Supplementary Material 1.** Script and questions.
(DOCX)

## Acknowledgments

We thank all the research participants who agreed to take part in the interviews and shared their perceptions of home-based diagnostic imaging services.

## Author contributions

**Conceptualization:** Lorena Jorge Lorenzi, Giovana Fondato Costa, Helianthe Kort, Paula Costa Castro.

**Data curation:** Lorena Jorge Lorenzi.

**Formal analysis:** Lorena Jorge Lorenzi, Giovana Fondato Costa, Helianthe Kort, Paula Costa Castro.

**Funding acquisition:** Lorena Jorge Lorenzi, Helianthe Kort, Paula Costa Castro.

**Investigation:** Lorena Jorge Lorenzi, Giovana Fondato Costa, Helianthe Kort.

**Methodology:** Lorena Jorge Lorenzi, Giovana Fondato Costa, Helianthe Kort, Paula Costa Castro.

**Project administration:** Lorena Jorge Lorenzi, Paula Costa Castro.

**Resources:** Lorena Jorge Lorenzi.

**Software:** Lorena Jorge Lorenzi, Giovana Fondato Costa.

**Supervision:** Helianthe Kort, Paula Costa Castro.

**Validation:** Lorena Jorge Lorenzi, Giovana Fondato Costa, Helianthe Kort, Paula Costa Castro.

**Visualization:** Lorena Jorge Lorenzi, Giovana Fondato Costa, Helianthe Kort, Paula Costa Castro.

**Writing – original draft:** Lorena Jorge Lorenzi, Giovana Fondato Costa.

**Writing – review & editing:** Lorena Jorge Lorenzi, Giovana Fondato Costa, Helianthe Kort, Paula Costa Castro.

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
