## [Decision Letter · Decision Letter 0]

21 Nov 2025

Dear Dr. Kort,

Thank you for submitting your manuscript to PLOS ONE. After careful consideration, we feel that it has merit but does not fully meet PLOS ONE’s publication criteria as it currently stands. Therefore, we invite you to submit a revised version of the manuscript that addresses the points raised during the review process.

We look forward to receiving your revised manuscript.

Kind regards,

Daniele Ugo Tari, M.D.

Academic Editor

PLOS ONE

“This study was financed, in part, by the São Paulo Research Foundation (FAPESP), Brasil. Process Number 2023/05218-6. The study was also financed by the Coordination for the Improvement of Higher Education Personnel - Code 001.”

“This study was financed, in part, by the São Paulo Research Foundation (FAPESP), Brasil. Process Number 2023/05218-6. The study was also financed by the Coordination for the Improvement of Higher Education Personnel - Code 001.”

“This study was financed, in part, by the São Paulo Research Foundation (FAPESP), Brasil. Process Number 2023/05218-6. The study was also financed by the Coordination for the Improvement of Higher Education Personnel - Code 001.”

4. We note that Figure 2 in your submission contain copyrighted images. All PLOS content is published under the Creative Commons Attribution License (CC BY 4.0), which means that the manuscript, images, and Supporting Information files will be freely available online, and any third party is permitted to access, download, copy, distribute, and use these materials in any way, even commercially, with proper attribution. For more information, see our copyright guidelines: http://journals.plos.org/plosone/s/licenses-and-copyright.

1. You may seek permission from the original copyright holder of Figure 2 to publish the content specifically under the CC BY 4.0 license.

Additional Editor Comments (if provided):

Dear Authors,

Your work is very interesting. Nevertheless, as highlighted by the reviewers, it needs several changes to be accepted for publication.

In particular, you should provide more details about the background of this study and the socio-economic characteristics of the population and the context in which it has been developed.

Furthermore, please, revise figure 2. It is not clear and it should be presented in a different way.

Please, revise the paper according to reviewers' suggestions.

Sincerely,

Reviewers' comments:

Reviewer's Responses to Questions

**Comments to the Author**

1. Is the manuscript technically sound, and do the data support the conclusions?

Reviewer #1: Partly

Reviewer #2: Partly

2. Has the statistical analysis been performed appropriately and rigorously?

Reviewer #1: N/A

Reviewer #2: No

3. Have the authors made all data underlying the findings in their manuscript fully available?

Reviewer #1: No

Reviewer #2: Yes

4. Is the manuscript presented in an intelligible fashion and written in standard English?

Reviewer #1: Yes

Reviewer #2: Yes

Reviewer #1: The authors give an interesting study regarding assessing the perceptions of patients who underwent home-based imaging diagnostics.

However, the authors begin the introduction by mentioning a key element of their study: diagnosis. Interestingly, they do not provide a rigorous definition of this concept, given that diagnosis appears to be the objective of the procedures they analyze. They should devote some space to this in the article, especially by defining what diagnosis means conceptually for the authors. The same is true when, also in the introduction, they use the word access. Given the structure of the article, the idea of access to health is always present. Again, access as a concept is neglected in the face of the large amount of data provided by the authors. Balancing the article by summarizing relevant data and incorporating crucial theoretical elements would improve it.

On the other hand, the authors work on patients' perceptions, but there is no theoretical literature associated with representation as a concept. In this sense, the idea of perceptions is confused with that of representations about health and the processes of health/illness/care. Regarding the latter concept (care), the authors must develop this idea further and explain how it is intertwined with the results and testimonies they present. In this sense, the conclusions appear compromised because they are not conceptually based on a sufficiently articulated theoretical framework.

It is unclear, or at least I did not see, whether the Unified Health System (SUS) covers the cost of these health examinations or if users must pay for their health plans privately to access these examinations. This is a key aspect to consider when thinking about perceptions (representations) in patients with home care services, regardless of their nature.

That said, the article has great potential and joins other works that attempt to understand the logic of home care in a broad sense.

Reviewer #2: Dear Authors

While it is interesting to qualify the use of imaging exams at home, I recommend the following major concerns to be considered. The concerns is in-hand, but it it is critical.

1-) The manuscript title indicates Home-Based "Imaging", while the manuscript -including results in table 2- mention bio signal measurements at home such as ECG and EEG. Therefore, I highly recommend that the manuscript is revised and then restricted to home based imaging exams only. It seems they are only four (X-RAY, USG, Echo, Doppler), not "eight" as you claimed in line 690-691 !

2-) Lines 573-575: it is really strange to present a word cloud with their sizes refer to their number of participants. Any alternative statistical presentation is recommended. Additionally, how this figure was generated, what statistics platform was used? please, explain the statistical platform, unless overlooked by me.

3-) why did you prefer analyzing results as Yes/No. For instance there are other statistics for illustrating survey such as likert. Please, discuss.

4-) Some references are not related to home based "imaging" such as reference 17. Restricting references to home based imaging or general home based health care is required.

In summary, please revise the manuscript and make it specialized to home-based imaging exams only, as stated in the title. I think that this matter is in-hand, so the above major revision/concerns are expected.

Best wishes

**Do you want your identity to be public for this peer review?** For information about this choice, including consent withdrawal, please see our Privacy Policy

Reviewer #1: No

Reviewer #2: No

---

## [Author Response · Author response to Decision Letter 1]

16 Jan 2026

Dear Editor and Reviewers,

We sincerely appreciate your valuable time and constructive feedback on our manuscript entitled “Patients' Perceptions of Home-Based Imaging Diagnostics: A Qualitative Study” (PONE-D-25-24934). We have carefully considered all the comments and revised the manuscript accordingly. Below, we provide a detailed, point-by-point response to each comment. All changes made in the revised manuscript are highlighted in yellow.

Editor(s)’ Comments to Author:

Comment 1: Please ensure that your manuscript meets PLOS ONE's style requirements, including those for file naming.

Response 1: Thank you for the comment, we have reviewed the guidelines and we confirm that the manuscript complies with PLOS ONE’s style requirements, including the file-naming directives.

Comment 2: Thank you for stating the following financial disclosure:

“This study was financed, in part, by the São Paulo Research Foundation (FAPESP), Brasil. Process Number 2023/05218-6. The study was also financed by the Coordination for the Improvement of Higher Education Personnel - Code 001.”

Response 2: The funders had no role in study design, data collection and analysis, decision to publish, or preparation of the manuscript. We have included this statement in the cover letter as requested.

Comment 3: Thank you for stating the following in the Acknowledgments Section of your manuscript:

“This study was financed, in part, by the São Paulo Research Foundation (FAPESP), Brasil. Process Number 2023/05218-6. The study was also financed by the Coordination for the Improvement of Higher Education Personnel - Code 001.”

“This study was financed, in part, by the São Paulo Research Foundation (FAPESP), Brasil. Process Number 2023/05218-6. The study was also financed by the Coordination for the Improvement of Higher Education Personnel - Code 001.”

Response 3: Thank you for your suggestion. The funding information currently provided in the system is correct. However, we will remove all funding-related content from the Acknowledgments section as requested. Accordingly, the Acknowledgments will be updated to: “We thank all the research participants who agreed to take part in the interviews and shared their perceptions of home-based diagnostic imaging services.”

Comment 4: We note that Figure 2 in your submission contain copyrighted images. All PLOS content is published under the Creative Commons Attribution License (CC BY 4.0), which means that the manuscript, images, and Supporting Information files will be freely available online, and any third party is permitted to access, download, copy, distribute, and use these materials in any way, even commercially, with proper attribution. For more information, see our copyright guidelines: http://journals.plos.org/plosone/s/licenses-and-copyright.

We require you to either (1) present written permission from the copyright holder to publish these figures specifically under the CC BY 4.0 license, or (2) remove the figures from your submission

Response 4: Thank you for this helpful comment. We removed Figure 2 from the manuscript submission. This modification aligns with a suggestion already made by one of the reviewers.

Reviewer 1:

Comments to the Author:

Comment 1: The authors give an interesting study regarding assessing the perceptions of patients who underwent home-based imaging diagnostics.

Response 1: We appreciate your comment.

Comment 2: However, the authors begin the introduction by mentioning a key element of their study: diagnosis. Interestingly, they do not provide a rigorous definition of this concept, given that diagnosis appears to be the objective of the procedures they analyze. They should devote some space to this in the article, especially by defining what diagnosis means conceptually for the authors.

Response 2: Thank you for your suggestion. We agree with you that the introduction lacked a clear explanation of the diagnostic concept underlying our study. We have now added a definition of diagnosis, along with additional information to make the introduction more robust and theoretically grounded. Below, we include the section that was inserted into the manuscript about the diagnosis:

“Diagnosing a health condition is essential to guide appropriate treatment. According to the Centers for Disease Control and Prevention [1], diagnosis can be defined as “the act or process of identifying or determining the nature and cause of a disease or injury through evaluation of patient history, examination of a patient, and review of laboratory data.” Health diagnoses can be obtained through various methods and different types of equipment, and among the most commonly used are imaging exams such as X-ray, ultrasound, computed tomography, and magnetic resonance imaging. Imaging diagnoses can identify a wide variety of health conditions, including stroke, thyroid disorders, appendicitis, coronary artery disease, among others [2].“

Comment 3: Also in the introduction, they use the word access. Given the structure of the article, the idea of access to health is always present. Again, access as a concept is neglected in the face of the large amount of data provided by the authors. Balancing the article by summarizing relevant data and incorporating crucial theoretical elements would improve it.

Response 3: Thank you for this helpful comment. We appreciate your observation. We have now incorporated a conceptual definition of access into the introduction, ensuring better balance between the empirical data and the theoretical foundations of the article. The added section is presented below:

“Access to health care is related to the opportunity to seek, obtain, and utilize health services, as well as to have the patient’s needs adequately met. Various factors may influence this access, including patient-specific characteristics, such as region of residence, socioeconomic status, and acceptability, as well as characteristics of the services, such as cost, location, adequacy, among others [5].”

Comment 4: On the other hand, the authors work on patients' perceptions, but there is no theoretical literature associated with representation as a concept. In this sense, the idea of perceptions is confused with that of representations about health and the processes of health/illness/care. Regarding the latter concept (care), the authors must develop this idea further and explain how it is intertwined with the results and testimonies they present.

Response 4: Thank you for your suggestion. We understand your concern. In response, we have expanded the introduction to clarify what is meant by patients’perceptions in the context of this study. Specifically, the perceptions analyzed refer to experiences related to receiving diagnostic imaging services at home. This includes aspects of service quality, as well as the views and impressions of individuals who underwent home-based imaging diagnoses. The study did not investigate perceptions regarding any specific disease or a single diagnostic category; instead, it focused broadly on the home-based diagnostic imaging service, which encompasses various types of exams performed at home, such as X-rays and ultrasounds. Additionally, Human-Centered Design was used as the theoretical framework to identify patients’ perceptions. Below, we provide the paragraph that has been added to the introduction and the methods:

“Therefore, these perceptions must be carefully assessed, as understanding how patients evaluate the service is fundamental for identifying the aspects they consider most valuable. Such insights not only guide institutional improvements and allow verification of ongoing enhancements but also contribute to the provision of patient-centered care [15].

To understand patients’ perceptions, it is important to consider Human-Centered Design, in which the focus of inquiry guiding the development of a service is the human being and their perspectives, with the aim of producing solutions that are truly usable. This approach requires identifying users’ needs and examining how they interact with the service. In the healthcare field, it involves the active participation of patients and is essential for the development of health services that are human-centered and aligned with the real-life contexts of end users [16].”

“The perceptions examined in this study refer to participants’ experiences with receiving diagnostic imaging services in the home setting. These perceptions encompassed aspects related to service quality, as well as the views and impressions of individuals who underwent home-based imaging procedures. The study did not explore perceptions associated with any particular disease or with a single diagnostic category. Instead, it adopted a comprehensive approach, focusing on the home-based diagnostic imaging service as a whole, which includes various types of examinations performed at home, such as X-rays and ultrasounds.”

Comment 5: It is unclear, or at least I did not see, whether the Unified Health System (SUS) covers the cost of these health examinations or if users must pay for their health plans privately to access these examinations. This is a key aspect to consider when thinking about perceptions (representations) in patients with home care services, regardless of their nature.

Response 5: Thank you for this helpful comment. In fact, the Unified Health System (SUS) to date does not finance or provide coverage for this examination. In most circumstances, the procedure is commissioned by private health insurance plans, which contract the service on behalf of their beneficiaries. In the other cases, when the examination is obtained privately, the health plan does not assume responsibility for its cost. In the case of the participants in our study, as all of them contracted the service privately, they were responsible for covering the cost themselves. We have added an explanation of this point to the results to ensure greater clarity.

“The Brazilian public Unified Health System (SUS) to date does not provide coverage for this examination. Generally, the procedure is commissioned by private health insurance plans, and when obtained on a private basis, its cost is not reimbursed. In the context of this study, all participants procured the service privately and, consequently, assumed full financial responsibility for the examination.”

Comment 6: That said, the article has great potential and joins other works that attempt to understand the logic of home care in a broad sense.

Response 6: We appreciate the reviewer’s positive assessment. We are pleased that the article is recognized as having strong potential and aligning with broader efforts to understand the logic of home care. All of the reviewer’s suggestions were valuable and have been fully incorporated into the revised manuscript.

Reviewer 2:

Comments to the Author:

Comment 1: The manuscript title indicates Home-Based "Imaging", while the manuscript -including results in table 2- mention bio signal measurements at home such as ECG and EEG. Therefore, I highly recommend that the manuscript is revised and then restricted to home based imaging exams only. It seems they are only four (X-RAY, USG, Echo, Doppler), not "eight" as you claimed in line 690-691!

Response 1: Thank you for this helpful comment. We agree that the manuscript title and scope required better alignment. Indeed, some of the technologies mentioned in the original version included biosignal measurements (such as ECG and EEG), which are not imaging techniques and therefore extended beyond the intended focus of the study.

In response we have implemented the following revisions:

● We restructured the manuscript to include only home-based imaging examinations.

● We removed ECG, EEG, and others from all sections, including Table 2.

● We revised the title, abstract, and introduction to accurately reflect the scope restricted to home-based imaging modalities.

After reassessing the content, we confirmed that the correct number of applicable home-based imaging modalities is four, and we corrected the statement that previously referred to “eight.”

In the response regarding the type of diagnostic imaging exams performed, we reported the other four (which are not imaging exams) simply as other types of tests performed at home. And throughout the entire manuscript, we removed information about these four.

We sincerely appreciate this comment, which has helped improve the accuracy and conceptual consistency of the manuscript.

Comment 2: Lines 573-575: it is really strange to present a word cloud with their sizes refer to their number of participants. Any alternative statistical presentation is recommended. Additionally, how this figure was generated, what statistics platform was used? please, explain the statistical platform, unless overlooked by me.

Response 2: Thank you for your suggestion. We agree that the use of a word cloud to represent the number of participants may appear unusual and is not the most appropriate statistical visualization for this type of data.

In response, we have taken the following steps:

● We removed the word cloud from the revised manuscript and now, there is just a table content explaining the subcategories found according to the participants' perceptions. This issue was also raised as a concern by the editor regarding the copyright of image 2, so we decided to remove it from the manuscript. However, this change did not cause any harm to the manuscript because all the data are presented in the table.

● We explained in the revised text that the table was generated based on the results of the qualitative data analysis conducted using NVivo software.

We thank the reviewer for pointing out this issue, which has improved the clarity and rigor of our data presentation.

Comment 3: why did you prefer analyzing results as Yes/No. For instance there are other statistics for illustrating survey such as likert. Please, discuss.

Response 3: Thank you for the comment. We thank the reviewer for raising this important point. As described in the methods section, the present study is a qualitative interview study. Our goal was to capture how participants experienced the services, highlighting positive and negative aspects, perceptions of reliability, preferences, preparation, and perceived quality of care.

During the inductive qualitative analysis, the themes emerging from participants’ narratives naturally aligned with the presence or absence of specific perceptions, attitudes, or concerns. Because the study did not use structured questionnaires or rating scales, but rather open-ended interviews, the data did not include graded or ordinal responses that would support the application of Likert-type analyses. Instead, categorizing the qualitative findings into Yes/No descriptors (e.g., “expressed concern” vs. “did not express concern”) provided a transparent and consistent way to summarize the frequency with which key themes appeared across interviews.

We acknowledge, however, that Likert scales and other quantitative measures can indeed offer richer granularity in survey-based research. We have now added a statement in the discussion noting that future studies employing structured questionnaires or

---

## [Decision Letter · Decision Letter 1]

9 Feb 2026

Patients' Perceptions of Home-Based Imaging Diagnostics: A Qualitative Study

PONE-D-25-24934R1

Dear Dr. Kort,

We’re pleased to inform you that your manuscript has been judged scientifically suitable for publication and will be formally accepted for publication once it meets all outstanding technical requirements.

Kind regards,

Daniele Ugo Tari, M.D.

Academic Editor

PLOS One

Additional Editor Comments (optional):

Reviewers' comments:

Reviewer's Responses to Questions

**Comments to the Author**

Reviewer #1: All comments have been addressed

Reviewer #2: All comments have been addressed

2. Is the manuscript technically sound, and do the data support the conclusions?

Reviewer #1: Yes

Reviewer #2: Yes

3. Has the statistical analysis been performed appropriately and rigorously?

Reviewer #1: N/A

Reviewer #2: Yes

4. Have the authors made all data underlying the findings in their manuscript fully available?

Reviewer #1: Yes

Reviewer #2: No

5. Is the manuscript presented in an intelligible fashion and written in standard English?

Reviewer #1: Yes

Reviewer #2: Yes

Reviewer #1: The authors managed to resolve the questions raised by this reviewer. I believe that the text is ready for publication, subject only to the editor's criteria.

Reviewer #2: (No Response)

**Do you want your identity to be public for this peer review?** For information about this choice, including consent withdrawal, please see our Privacy Policy

Reviewer #1: **Yes:** Ramiro Andres Fernandez Unsain

Reviewer #2: **Yes:** Abdel-Razzak Al-Hinnawi

---

## [Editor Report · Acceptance letter]

PONE-D-25-24934R1

PLOS One

Dear Dr. Kort,

I'm pleased to inform you that your manuscript has been deemed suitable for publication in PLOS One. Congratulations! Your manuscript is now being handed over to our production team.

Kind regards,

on behalf of

Dr. Daniele Ugo Tari

Academic Editor

PLOS One